# FigCaps-HF: A Figure-to-Caption Generative Framework and Benchmark with Human Feedback

**Benchmark:** `https://figshare.com/s/c034fd77bea9475319cb`
**Code:** `https://github.com/FigCapsHF/FigCapsHF`
**Documentation:** `https://figcapshf.github.io/`

## Abstract

Captions are crucial for understanding scientific visualizations and documents. Existing captioning methods for scientific figures rely on figure-caption pairs extracted from documents for training, many of which fall short with respect to metrics like helpfulness, explainability, and visual-descriptiveness (Huang et al., 2023) leading to generated captions being misaligned with reader preferences. To enable the generation of high-quality figure captions, we introduce **FigCaps-HF** a new framework for figure-caption generation that can incorporate domain expert feedback in generating captions optimized for reader preferences. Our framework comprises of 1) an automatic method for evaluating the quality of figure-caption pairs, and 2) a novel reinforcement learning with human feedback (RLHF) method to optimize a generative figure-to-caption model for reader preferences. We demonstrate the effectiveness of our simple learning framework by improving performance over standard fine-tuning across different types of models. In particular, when using BLIP as the base model, our RLHF framework achieves a mean gain of 35.7%, 16.9%, and 9% in ROUGE, BLEU, and Meteor, respectively. Finally, we release a large-scale benchmark dataset with human feedback on figure-caption pairs to enable further evaluation and development of RLHF techniques for this problem.

## 1 Introduction

For scientific articles, figures like graphs, charts and plots are key to effectively conveying the work's motivation, methodology, and results to readers. To better understand a given figure and, by extension, the research work itself, it is then crucial that the corresponding captions are informative, i.e., a given caption can represent and complement the figure, situating it in the context of the article. While the importance of figure captions is universally acknowledged, writing a good caption is not trivial. More often than not, many scholarly works contain generic figure captions and lack descriptiveness, thus rendering the figure unhelpful. This has motivated extensive research into developing methods that can automatically generate captions for figures to assist researchers in writing better captions.

Recent works in figure-captioning formulate the problem as a vision-to-language task and have primarily focused on developing methods to encode the figure image and metadata and decode captions effectively. For model training, these methods use figure-caption pairs extracted from existing scientific articles (Hsu et al., 2021). While this method of data collection is appealing due to its easy access, this also leads to the problem of poor model learning and generalization when the captions are not well written. As discussed in Huang et al. (2023), more than 50% of the captions in arXiv cs.CL papers were classified as not helpful to the domain expert readers. Thus, figure-captioning methods trained on such data are not calibrated to reader preferences and thus generate captions that are uninformative.

Motivated by the above, we introduce **FigCaps-HF**, a new benchmark and learning framework for improving figure-to-caption generation by aligning model optimization to reader preferences. Figure 1

describes our proposed framework, designed around two key questions: **(1)** How can we incorporate feedback from domain experts in a computationally efficient manner without compromising on performance and usability? **(2)** How can we develop a scalable framework for feedback generation that minimizes human labeling efforts?

To address **(1)** we utilize offline Upside-Down RL (⅂Я or UDRL) to align the model's generated captions with expert feedback. Unlike previous applications of RLHF (Ouyang et al., 2022) which uses on-policy algorithms (Schulman et al., 2017) for reward maximization, our approach of using offline reward-conditioned behavioral cloning for model optimization is computationally efficient. Once our reward model is trained and we predict the reward scores for each sample, we do not need the reward model during figure-to-caption model training. Furthermore, offline UDRL-like methods are known to perform equally well as their other counterparts (Emmons et al., 2021) while being efficient and simple.

To address **(2)** we introduce a general caption-scoring mechanism, guided by domain expert feedback, which allows us to evaluate quality of figure-caption pairs with respect to readers preference. Specifically, we utilize a small human-annotated dataset of image-caption pairs, each rated on a variety of factors, e.g., helpfulness, OCR content, takeaway etc, to train an auxiliary model to score for a given caption on the basis of the quality measure. This step is integral because it allows us to infer caption scores for our larger training set. Additionally, we publicly release our benchmark dataset with feedback for future research on developing figure-to-caption models.

Our experimental results show an increase in performance by using our Upside-Down RL-guided approach. Firstly, our empirical results indicate that our trained reward model is very well calibrated, and the annotation statistics of our ground-truth annotations match those from our inferred annotations. Secondly, we evaluate the performance of our approach on a variety of image-to-text models and observe that models with RLHF achieve the best performance; specifically, our best-performing model has a 35.7% increase in BLEU, 16.9% increase in ROUGE-L, and 9% increase in METEOR score using RLHF. Our ablation studies show the beneficial effects of further investigation into parts of our setup, including the type and nature of feedback used.

**Summary of main contributions.** We summarize the key contributions of this work as follows:

- We introduce an RLHF benchmark and framework for figure-to-caption generation that leverages a *small amount* of actual human feedback to learn an oracle model to infer human feedback on a larger scale for any unknown figure-caption pair encountered in the wild.

- We propose a technique that learns an oracle model from a small amount of human feedback, which can then be used for predicting the human feedback scores for any new unseen figure-caption pair.

- Extensive experiments demonstrate the effectiveness of our benchmark and framework for figure-to-caption generation via human feedback.

- To facilitate further research on this important new problem, we release a comprehensive benchmark dataset for figure-to-caption generative models with human feedback. This new benchmark will enable the research community to develop even better RLHF models for figure-to-caption generation.

## 2 BACKGROUND

**Figure Caption Generation.** Most prior work in scientific figure-captioning can be broadly divided into the following three categories based on their different input modalities: the figure-image alone, the underlying data chart of the figure, and relevant texts from the original article. In the vision-based approach, prior works have primarily utilized a vision-encoder to encode figure-features followed by a text-decoder to generate captions. Siegel et al. (2016); Qian et al. (2021; 2020) focus on explicitly extracting different features of the figure before combining their information for downstream tasks. Chen et al. (2019; 2020a;b) create and leverage FigCAP, a synthetic figure-caption dataset and adapt an LSTM model to produce captions. More recently, Hsu et al. (2021) collected a dataset, namely SciCAP, from published articles and used a CNN+LSTM pipeline to generate captions. There are few prior works which examine the abilities of utilizing SOTA image-captioning pipelines, which primarily utilize large pre-trained Transformer (Vaswani et al., 2017) modules, for figure-captioning.

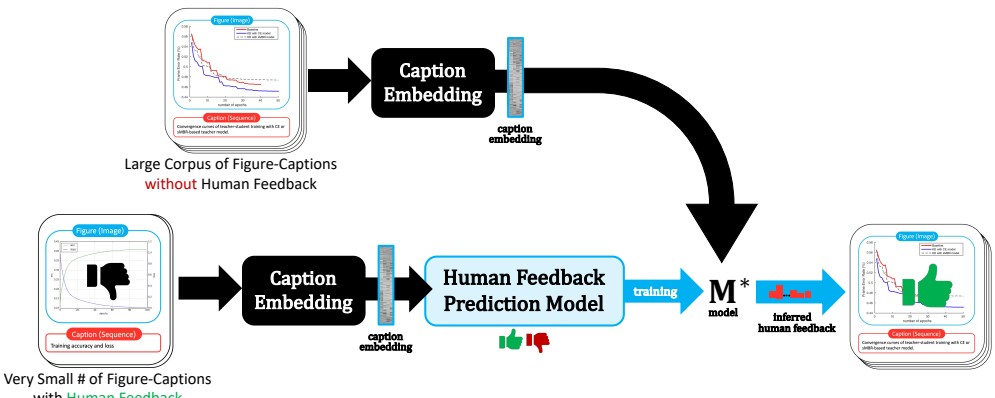

Figure 1: RLHF Framework for Figure-Caption Generative Models

A closely related task is Figure Question Answering, which formulates the more general problem of figure understanding as a visual-question answering task; there has been a variety of works in this space towards modeling (Siegel et al., 2016; Kahou et al., 2017; Li et al., 2022b; Singh & Shekhar, 2020; Zou et al., 2020; Kafle et al., 2018; 2020) as well as creating curated datasets including DVQA (Kafle et al., 2018), FigureQA (Kahou et al., 2017), PlotQA (Methani et al., 2020), Leaf-QA (Chaudhry et al., 2020), and ChartQA (Masry et al., 2022). In the data-driven approach, research focuses on using only the tabular data, as well as some metadata, to generate a caption. Table-to-Text(Yin et al., 2019) focuses on generating captions for rows in arbitrary tables. Chart-to-Text (Obeid & Hoque, 2020) creates a new large-scale dataset focusing on figure-captioning and adopt an encoder-decoder transformer model to process the underlying data table and generate summaries. In the text-driven approach, Huang et al. (2023) focuses on utilizing only the relevant text in an article to generate a figure-caption, for example, using text explicitly referencing the figure.

**Learning with Human Feedback** Aligning model predictions with human preference has been shown to improve task performance in various areas, including natural language processing tasks like language model pretraining (Korbak et al., 2023), machine translation (Bahdanau et al., 2016; Kreutzer et al., 2018), text summarization (Stiennon et al., 2020), unlearning undesirable behaviors from language models (Lu et al., 2022), computer vision tasks like text-to-image generation (Lee et al., 2023; Zhang et al., 2023) and reinforcement learning tasks like training RL agents (MacGlashan et al., 2017; Ibarz et al., 2018; Lee et al., 2021). In contrast to prior works, our work also aims at improving figure caption generation by optimizing model learning to align with domain expert feedback. However, unlike previous work that leverages on-policy RL (Schulman et al., 2017) algorithm to maximize the reward-weighted likelihood, our framework utilizes reward-conditioned behavioral cloning (Emmons et al., 2021), an offline variant of upside-down RL method (Srivastava et al., 2019) to optimize model learning for reader preference. This provides a simpler and more controllable framework for human preference alignment. Furthermore, our feedback scheme allows for incorporating multiple feedback at different granularity as reward signal during the model optimization step, thus improving model learning. We propose a general human-feedback model along with a new benchmark with feedback to enable further research in developing and evaluating methods that optimize for reader preference.

## 3 FRAMEWORK

In this section, we explain our proposed framework for learning with expert feedback (Figure 1). We first describe a standard figure-captioning pipeline (Sec. 3.1). Next, we provide details of designing and training a generalizable human-feedback prediction model (Sec. 3.2). Finally, we discuss our feedback-aligned model training strategy instantiated as a simple RLHF framework (Sec. 3.3).

### 3.1 PRELIMINARIES

In a figure-captioning problem, we are initially provided with a dataset $D_w$ consisting of figure-caption pairs $\{I_w, T_w\}$. Given the dataset $D_w$, we can then define a model $f_\theta$, that takes in information

corresponding to the figure and outputs a sequence of text as output. Typically, the input consists of only the figure image. However, other sources of information like figure-specific text from the corresponding document, OCR, figure metadata can also be utilized as input samples.

Assuming the general case of figure image as input, model $f_\theta$ is constructed using a vision encoder module to get image-based encoding and a language encoder-decoder module to encode and generate corresponding text. The weights $\theta$ can either be randomly initialized, or initialized by large-scale pretrained model weights. Furthermore, the model weights corresponding to the vision encoder and text encoder-decoder models can either be initialized with separate weights or jointly trained model weights. After initialization, model $f_\theta$ can then be trained for the task of caption generation.

Generally, for training such a model, Language Modeling (LM) loss is used as a standard training objective. Let $\{I_i, T_i\} \in D$ be the input to the model $f_\theta$, where $I_i \in \mathbb{R}^n$ is the input figure, and $T_i$ is the corresponding text sequence. Additionally, $T_i$ is represented as sequence of $K_j$ tokens from a fixed vocabulary $\mathcal{V}$: $T_i = (T_{i,1}, ...T_{i,K_j})$, where $K_j = |T_i|$. Then the training objective is defined as:

$$\mathcal{L}_{\text{LM}} = \frac{1}{K_j + 1} \sum_{j=0}^{K_j+1} H(T_{i,j}|I_i, (T_{i,0}, ..., T_{i,j-1})),\qquad(1)$$

where H denotes the cross-entropy loss and $(T_{i,0}, ..., T_{i,j-1})$ represents all the tokens in the caption prior to $T_{i,j}$.

## 3.2 HUMAN FEEDBACK PREDICTION MODEL

To improve figure-to-caption generation, we propose to incorporate domain expert feedback into our optimization step. To generate feedback for figure-caption pairs, we thus propose to learn a feedback prediction model to score individual datasample based on different metrics representing reader preferences. Our objective is to learn a model that can predict human feedback scores for unseen captions accurately given small set of training samples.

To this end, we first label a small control set $D_h$ consisting of $M$ figure caption pairs $\{I_w, T_w\}$ with domain experts ratings. Here we assume that $M \ll N$, i.e. the size of the control set is significantly less than the original noisy dataset (For example, if $N = 100,000$, then $M = 100$). We can now train a model on $D_h$ to predict the human expert ratings for the original dataset $D_w$. Specifically, given human feedback dataset $D_h$ containing figure-caption pairs $\{I_h, T_h\} \in D_h$ and $k$ human expert evaluation metrics for each datasample $y \in y_0, y_1, ...y_k$, we want to train $k$ models $R(x_i, \theta)_k$ to predict the $k$ scores respectively. Here the output of a model $R(x_i, \theta)_k(T_h)$ is a scalar quantity denoting a specific metric score for the given input caption. Thus we formulate the scoring problem as a regression task. Specifically, we can define our human-feedback prediction model as follows:

$$R(x_i, \theta)_k(T_h) = g(l(\theta_l, x_i), \theta_g),\qquad(2)$$

where, $R(x_i, \theta) : \mathbb{R}^N \to \mathbb{R}$, $l(x_i, \theta_l) : \mathbb{R}^N \to \mathbb{R}^D$ and $g(u_i, \theta_g) : \mathbb{R}^D \to \mathbb{R}$. In the above, $l(., \theta_l)$ is an embedding function that takes in input data $x_i \in \mathbb{R}^N$ and generates corresponding representation $u_i \in \mathbb{R}^D$, and $g(., \theta_l)$ is a regression function to generate the scores respectively. We only train the regression function while keeping the weights of the embedding function fixed. For training the regression function, we use mean-squared error loss, written as: $\mathcal{L}_{\text{R}} = \sum_{i=1}^{D_h} (\hat{y}_i - y_i)^2$, where $\hat{y}_i$ is the predicted score while $y_i$ is the ground-truth evaluation score. After training the human-feedback prediction models, we compute scores for all the samples in the training dataset $D_w$ to construct our new set, which will be used for training the figure-caption model.

## 3.3 REINFORCEMENT LEARNING WITH HUMAN FEEDBACK

Given the human-feedback prediction model described above, we can now use it as a reward model to train an image-to-text model that generates higher-quality captions. We achieve this goal, by formulating the problem as a reinforcement learning task. Specifically, for the given training dataset $D_w$ containing figure caption pairs $\{I_w, T_w\}$, we can consider figures $I_w$ as the state of the environment, caption $T_w$ as the actions and the corresponding predicted metric scores $R(T_w)$ for these captions as the rewards/outcomes. Then our objective is to learn a policy (which in this case would be the image-to-text model $f(\theta)$ that we want to train) that maps from states($I_w$) to actions($T_w$)

| | # Fig-Caption Pairs | Human Feedback | Median | Mean | Std | Q1 | Q3 |
|---|---|---|---|---|---|---|---|
| ACTUAL HUMAN FEEDBACK | **438** | **Helpfulness** | 3 | 3.01 | 1.19 | 2 | 3 |
| | | **Takeaway** | 2 | 2.16 | 1.22 | 1 | 2 |
| | | **Visual** | 2 | 2.11 | 1.08 | 1 | 2 |
| | | **OCR** | 4 | 3.83 | 0.80 | 4 | 4 |
| PREDICTED HUMAN FEEDBACK | **106,834** | **Helpfulness** | 2.89 | 2.89 | 1.07 | 2.17 | 3.61 |
| | | **Takeaway** | 1.95 | 2.06 | 1.03 | 1.33 | 2.66 |
| | | **Visual** | 1.91 | 2.02 | 1.01 | 1.31 | 2.63 |
| | | **OCR** | 3.88 | 3.84 | 0.83 | 3.32 | 4.41 |

Table 1: Summary of our benchmark dataset for figure-caption generative models with RLHF.

such that we maximize the reward for each action. In this way, we can generate output captions that better align with human judgment of a good figure-caption.

While there are many different approaches in the reinforcement learning literature (Schulman et al., 2017) to achieve the above objective, we specifically focus on offline upside-down reinforcement learning (UDRL). We select offline UDRL because it computationally efficient and robustly performant without being algorithmically complex (Emmons et al., 2021). In UDRL, the motivation is to learn a policy ($\pi_\theta$) that maps the states ($S_t$) to actions ($a_t$) conditioned on specific rewards($r_t$). Thus the learning problem can be formulated as a supervised learning problem, wherein we first sample the triplets of $S_t, a_t, r_t$ from the environment to construct our dataset, which is then used to train $\pi_\theta$ using standard supervised learning objective. Specifically, we can write the optimization problem as:

$$\max_\theta \sum_{t \in D} \mathbb{E}[\log \pi_\theta(a_t|S_t, r_t)], \tag{3}$$

We follow the above UDRL framework for learning an image-text model $f(\theta)$. For our setting, we consider our image-to-text model $f(\theta)$ as our policy $\pi_\theta$. For each caption $T_i \in T_w$, we compute a reward score and quantize it to generate a control token $c_i$. Specifically, we binarize the reward score to generate two control tokens: $<|\text{good}|>$ and $<|\text{bad}|>$. In general, the level of quantization is a hyperparameter which can be selected according to task or other factors. For each caption $T_i \in T_w$, we compute the control token by thresholding the output of $R$, i.e. if $R(I_i, T_i) \geq t$ then $c_i = <|\text{good}|>$, else $c_i = <|\text{bad}|>$. Here $t$ is a hyperparameter. Given the additional human feedback, we fine-tune $f_\theta$ with the following new objective function:

$$\mathcal{L}_{\text{HF}} = \frac{1}{K_j + 1} \sum_{j=0}^{K_j+1} H(T_{i,j}|I_i, (c_i, T_{i,0}, ..., T_{i,j-1})), \tag{4}$$

where $c_i$ refers to the control token computed using the reward function $R$ for a given caption $T_i$.

## 4 FIGCAPS-HF: FIGURE-CAPTIONING WITH HUMAN FEEDBACK BENCHMARK

As noted before, captions from online scientific articles can be of 'low quality' with respect to domain expert quality metrics (Huang et al., 2023). This can, in turn, lead to poor figure-captioning models as these are trained to simply maximize the likelihood of the raw training data. Thus, our goal with the new benchmark is to provide additional training signals to improve figure-caption model without incurring the cost of re-creating a new dataset.

To this end we propose our new benchmark for figure-captioning with feedback. Our benchmark consists of 133,543 figure-caption pairs (Hsu et al., 2021) with feedback scores. Our dataset contains feedback based on different measures to evaluate quality of the author written captions for the corresponding figure. For each figure-caption pair, we evaluate the data sample based on four quality measures: **(1) Helpfulness**, **(2) Takeaway**, **(3) Visual-descriptiveness (visual)** and **(4) Image-text (OCR)** (Huang et al., 2023). Each quality metric is selected to measure the ability of the readers to comprehend and draw inferences based on the provided figure and the corresponding caption.

| | MODEL | #Params | ROUGE-L | BLEU | METEOR |
|---|---|---|---|---|---|
| OCR-ONLY | **Pegasus** | 0.27B | 0.026 | 4.78e-4 | 0.042 |
| FIGURE-ONLY | **TrOCR** | 0.23B | 0.025 | <0.001 | 0.018 |
| | **BEiT+GPT2** | 0.24B | 0.142 | 0.005 | 0.124 |
| | **ViT + RoBERTA** | 0.23B | 0.140 | 0.012 | 0.121 |
| | **ViT + GPT2** | 0.24B | 0.142 | 0.018 | 0.126 |
| FIGURE-CAPTION | **PromptCap** | 0.47B | 0.130 | 0.009 | 0.082 |
| | **Flamingo** | 1.14B | 0.087 | 0.001 | 0.046 |
| | **GIT** | 0.17B | 0.119 | 0.002 | 0.091 |
| | **BLIP** | 0.25B | 0.130 | 0.014 | 0.132 |
| | **CLIPCap** | 0.15B | 0.103 | 0.012 | 0.131 |
| RLHF | **Ours-BLIP-RLHF** | 0.25B | **0.152** | 0.019 | **0.145** |
| | **Ours-ViT+GPT2-RLHF** | 0.24B | 0.138 | **0.020** | 0.126 |

Table 2: Comparison with state-of-the-art methods. For all the metrics, higher values are better (↑).

We compute the feedback scores for each data sample in a scalable manner by first annotating a small subset with domain-expert feedback and then predicting score for the entire dataset using the human-feedback model described in Sec. 3.2. Specifically, we select 438 randomly sampled figure-caption pairs, each annotated by domain experts (Huang et al., 2023). Each pair has been evaluated on 5-point Likert scale for each of the above mentioned quality metric. Using this labeled subset, we train a human-feedback prediction model to generate scores for the remainder of the dataset. Unlike the subset, we keep the scores for the entire dataset as a continuous value. This allows the users of the benchmark to accordingly decide their own scheme for labeling each figure-caption pair based on different thresholding criteria, thus providing flexibility for fine-grained feedback.

Table 3.3 presents an overview of the statistics related to the actual and predicted human feedback for the captioning of scientific figures. We see that the predicted human feedback values in our study show a diverse range, as indicated by the small standard deviation of $1 \pm 0.2$ and a consistent mean value across all ratings. Additionally, the alignment of the median predicted scores with the actual human feedback values indicates that the model's performance is not skewed towards any particular rating but provides an accurate assessment across the range of ratings. This suggests that the human-feedback prediction model used to infer the scores is generalizable and can accurately assess the quality of captions across various ratings. Furthermore, the proposed model provides reliable scores for captions that fall outside the typical range of scores. For further implementation details, please refer to the section "Additional Dataset Details" in the appendix.

## 5 EXPERIMENTS

**Setup.** For our human-feedback prediction model, we use MCSE (Zhang et al., 2022) as embedding function and a 2-layer MLP as regression function. For comparative evaluation, we select the following models as our baselines based on input: (1) OCR-only: Pegasus(Zhang et al., 2020), (2) Figure-only: TrOCR (Li et al., 2021), BeiT+GPT2, ViT+GPT2 (Dosovitskiy et al., 2021), ViT+RoBERTA (Dosovitskiy et al., 2021; Liu et al., 2019) and (3) Figure-Caption: PromptCap (Hu et al., 2022), Flamingo (Alayrac et al., 2022), GIT (Wang et al., 2022a), BLIP (Li et al., 2022a) and CLIPCap (Mokady et al., 2021). We use ROUGE-L (Lin, 2004), METEOR (Banerjee & Lavie, 2005) and BLEU (Papineni et al., 2002) metrics to compare each model's performance. For more details regarding individual baselines, metrics and dataset, please refer to the Appendix.

### 5.1 RESULTS

We show our experimental results in Table 2. Specifically, we want to evaluate the performance of our RLHF framework for figure-caption generation. To this end, we compare our framework with standard fine-tuning method and benchmark the performance on the Test set of our proposed

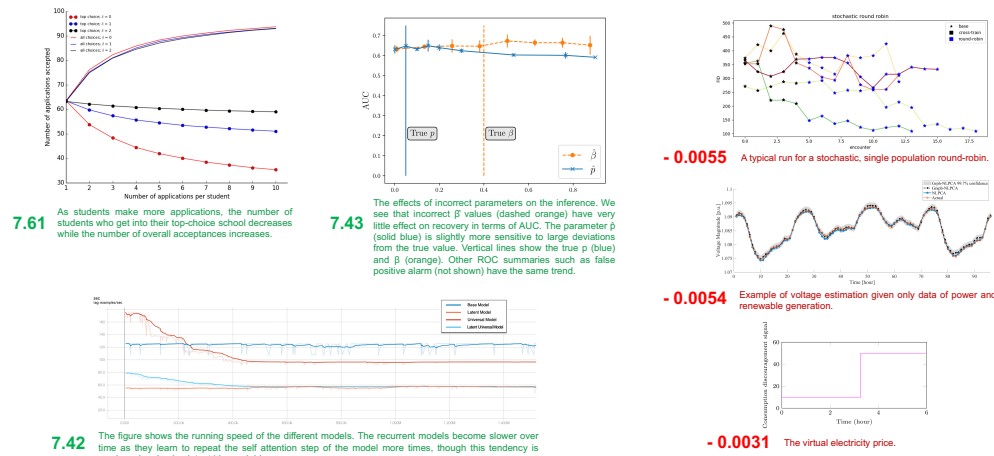

Figure 2: Results of our Human Feedback Prediction Model. Here we show the three figure-caption pairs with the highest (left; green) and smallest (right; red) "helpfulness" human feedback score from our trained HF model. Notably, the figure-caption pairs rated highly by our human-feedback predictive model are those that are obviously better as they mention specific takeaways, as well as OCR from the figure, and even visual aspects are often mentioned. In contrast, the figure-caption pairs with lowest scores by our predictive model are those that are extremely vague, without actual takeaways, OCR mentions, and without mentioning any visual aspects from the figure.

benchmark. We show fine-tuning results for all the above mentioned baselines. We use BLIP and ViT+GPT2 to evaluate our RLHF framework. From Table 2, models trained using our proposed RLHF formulation performs better than simple fine-tunning. Specifically, for BLIP, RLHF provides has a 35.7% increase in BLEU, 16.9% increase in ROUGE-L, and 9% increase in METEOR score. For ViT+GPT2, RLHF provides a 11.1% increase in BLEU.

Aggregating the metrics, BLIP performs best, which is likely due to its aligned image encoder and text decoder which are pre-trained jointly. In contrast, ViT+GPT2's modules are not aligned/trained jointly and the text decoder learns to attend to the vision encoder only during fine-tuning. Hence, for our approach, the type of pre-training can have an impact on the amount of model improvement.

Overall, since the performance increase is generalized among models with different pre-training strategies and overall model-structure, the results show the benefits of using this simple UDRL framework for fine-tuning. Utilizing only a small amount of human annotated data, different scoring mechanisms and prompts can be further developed to take advantage of this limited supervision and further increase performance.

## 5.2 QUALITATIVE RESULTS

To validate our frameworks ability to generate better reader-aligned captions than standard approaches, we conduct an extensive qualitative study. We evaluate the results of the human feedback prediction model and the figure-captioning models trained with RLHF. We provide our analysis below:

**Human Feedback Prediction Model**: To evaluate the generalizability our model, we first computed the score predictions on all the figure-caption pairs. Then we ordered the figure-caption pairs by the predicted scores and selected the top-3 figure-caption pairs with the largest score along with the bottom-3 figure-caption pairs with the smallest score. Results are provided in Figure 2. We observe that the figure-caption pairs with the largest scores are highly helpful to the reader as they mention specific takeaways from the figure (*e.g.*, "as students make more applications, the number of students who get into their top-choice school decreases, while the number of overall acceptances increases."), as well as mentioning specific visual aspects that are important to the understanding of it (*e.g.*, "... Vertical lines show the true p (blue) and $\beta$ (orange)"). In contrast, the bottom-3 figure-caption pairs scored the lowest (shown in red on the right in Figure 2) are vague, without any takeaways, nor reference to visual elements in the figure.

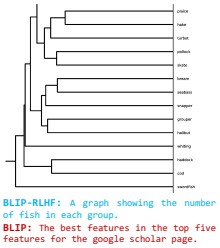
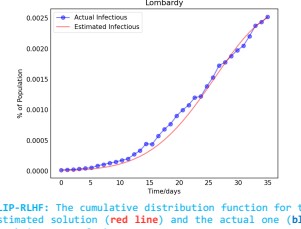

BLIP-RLHF: The average time of the training process and the baseline with different experiments.
BLIP: A dataset that has been generated by an agent, showing the number of edges that can be found in each experiment.

BLIP-RLHF: A graph showing the number of fish in each group.
BLIP: The best features in the top five features for the google scholar page.

BLIP-RLHF: The cumulative distribution function for the estimated solution (red line) and the actual one (blue dots) in terms of time.
BLIP: The IOT dataset.

Figure 3: Generated captions from our RLHF framework using BLIP as the base model compared to BLIP without RLHF.

| MODEL | ROUGE-L | BLEU | METEOR |
|---|---|---|---|
| **Binary Feedback** | 0.152 | 0.019 | 0.145 |
| **Multi-label Feedback** | 0.153 | **0.022** | **0.151** |
| **Binary + Multi-label Feedback** | **0.156** | 0.019 | 0.148 |

Table 3: Results with multi-labeled human feedback.

**Figure-Caption Generative Model**: To evaluate the quality of captions, we compare the output of BLIP-RLHF and BLIP (Fine-tuned) models. We show some of the results in Figure 3. In general we see that, qualitatively BLIP-RLHF produces better captions compared to fine-tuned BLIP. In most cases, captions produced by BLIP (Fine-tuned) are either explaining the given figure incorrectly (Figure 3, leftmost sub-figure), not relevant (Figure 3, middle sub-figure) or are completely uninformative (Figure 3, rightmost sub-figure). On the other hand, captions produced by BLIP-RLHF method are more faithful to the figure, captures semantic relation between texts to summarize the phenomenon and utilizes visual attributes in explaining the figure. We provide more examples and analysis in the Appendix.

## 5.3 ABLATION STUDY

We perform the following ablation experiments to better understand different components of our framework. We provide the details of our findings below.

**Effect of different human feedback labels**: To understand how the level of quantization of our reward signals (Binary vs Multi-level) affect the model learning, we conduct the comparative study by modifying the feedback while training the BLIP-RLHF model. First, we trained the model for 10 epochs using multi-labeled human feedback (Row 2), specifically, we used 5 levels of human feedback (very bad, bad, neutral, good, very good) calculated at the $20^{th}$, $40^{th}$, $60^{th}$, $80^{th}$ percentile respectively to ensure an equal number of samples. We also experimented with varying label coarsity during the course of training (Row 3); specifically, we trained the model with 5 epochs of binary-label feedback followed by 5 epochs of multi-label feedback. We show our results in Table 3. Both aforementioned approaches with finer feedback outperform simple binary feedback and demonstrate, through our RL framework, the model's ability and receptiveness to leverage more finer human feedback effectively. The experiment also indirectly validates the quality of our human prediction model, which is capable of providing useful labels at different levels of coarsity that can be leveraged

| | ROUGE-L | BLEU | METEOR |
|---|---|---|---|
| **Helpfulness** | 0.1520 | 0.0186 | 0.1450 |
| **Takeaway** | 0.1676 | 0.0230 | 0.1598 |
| **Visual** | 0.1678 | 0.0230 | 0.1595 |
| **OCR** | 0.1654 | 0.0223 | 0.1565 |

Table 4: Results with different human feedback metrics (BLIP-RLHF).

|          | ROUGE-L | BLEU    | METEOR |
|----------|---------|---------|--------|
| **BERT**   | 0.1565  | 0.01927 | 0.1473 |
| **SciBERT** | 0.1577  | 0.0201  | 0.1509 |
| **BLIP**   | 0.1573  | 0.01977 | 0.1494 |

Table 5: Results with different embedding models for the human-feedback model.

|             | MSE              |
|-------------|------------------|
| Helpfulness | $0.082 \pm 0.12$ |
| Visual      | $0.076 \pm 0.2$  |
| Takeaway    | $0.087 \pm 0.17$ |
| OCR         | $0.095 \pm 0.13$ |

Table 6: Evaluation of out-of-sample generalization

for increased performance on a downstream task like figure-captioning. The study also shows the further potential gains that can be made by further investigating different feedback mechanisms.

**Effect of different human feedback metrics**: We also study the effect of using different metrics as feedback for training the figure-caption models. In particular, we compare results of training the BLIP-RLHF model with the Helpfulness, Takeaway, Visual-descriptiveness (visual) and Image-text (OCR) feedback scores provided in our benchmark. We provide the results in Table 4. We see that training BLIP-RLHF with Takeaway, visual and COR feedback performs better than Helpfulness. This is understandable as helpfulness rating is subjective while Visual and Takeaway are objective evaluation metrics respectively. This shows that the type of feedback is important and that further gains can be made by modeling different aspects of the annotated human dataset.

**Effect of different figure-caption representations**: To understand the effect of using different figure-caption representations, we use BERT, SciBERT and BLIP to encode our figure-captions pairs and use their final-layer representations of the [CLS] token to train our human feedback prediction model. The results are provided in Table 5. The different representations outperform our default MCSE implementation, indicating that our human feedback prediction model, and downstream figure-captioning performance, are sensitive to the quality of representations used. Additionally, further performance gains can be made by using different representations, for example, by encoding different modalities (text only vs joint encoding of text and vision).

**Generalizability of the human feedback prediction model**: To evaluate the out-of-sample generalization of our human-feedback prediction model, we conduct a 5-fold cross-validation experiment on the original 438 annotated. We repeated the above experiment 5 times. We report our results (mean squared error (MSE) and standard deviation over 5 runs) in Table 6. As can be seen from the results, our model is able to achieve good results on the validation set. This highlights that our human-feedback prediction model demonstrates out-of-sample generalization and proves the statistical significance of our model.

| Training Size | MSE   | Gain   |
|---------------|-------|--------|
| 25% (109)     | 0.579 | 91.72% |
| 50% (219)     | 0.323 | 6.95%  |
| 100% (438)    | 0.311 | 2.98%  |
| 125% (657)    | 0.309 | 2.32%  |
| 200% (876)    | 0.302 | 0%     |

Table 7: Results varying the training size used for learning the human feedback prediction model (for inferring "Helpfulness"). Note gain is computed with respect to the best (lowest) MSE obtained (0.302). See text for detailed discussion.

| | MODEL | ROUGE-L | BLEU | METEOR |
|---|---|---|---|---|
| RLHF-APPEND | **Ours-BLIP-RLHF** | 0.136 | 0.018 | 0.132 |
| | **Ours-VIT-GPT2-RLHF** | 0.138 | 0.016 | 0.119 |
| RLHF-PREPEND | **Ours-BLIP-RLHF** | 0.152 | **0.019** | **0.145** |
| | **Ours-ViT+GPT2-RLHF** | 0.138 | **0.020** | 0.126 |

Table 8: Comparing RLHF prepend to append.

**Varying training size**: To evaluate the effectiveness of our approach when varying the number of samples used during training, we train the human feedback prediction model using 25%, 50%, 100%, 125%, and 200% of the human-annotated data. We used a held-out set of 300 samples for model evaluation of each of these models. We then trained separate models for each training set for the task of predicting the 'Helpfulness' measure. The results showing mean-squared error (MSE; lower is better) are provided in Table 7. Notably, we see the test performance of the model saturates as the number of training samples is increased. Even with 50% of the original human-annotated data, the model achieves good test results.

**Effect of human feedback position**: To understand the sensitivity of the model to the position of human feedback, we compare the performance of appending and pre-pending the human feedback labels in Table 8. Since our models generate text, during test time, without any human feedback label prompt, they can only rely on feedback during training. Additionally, due to the auto-regressive generation of our models, they only observe the label before generation, and for append, only observe the label after generation. Intuitively, pre-pending should work best since the generation is conditioned on the label. The results support this and show that ViT+GPT2 and BLIP perform better when trained with pre-pended human feedback.

## 6 CONCLUSION

In this work, we contribute a new benchmark and methodology to improve caption generation for scientific figures. We show that incorporating domain expert feedback in learning a model for figure-to-caption generation improves both model performance and caption quality. The proposed benchmark of figure-caption pairs with caption quality scores to further the research efforts in reader-aligned figure-captioning tasks. We hope that this new benchmark dataset will allow researchers to benchmark their own methods for incorporating human feedback in figure-to-caption generation tasks and various other image-to-text generation tasks. Future work will explore techniques to incorporate multiple complementary feedback as well as different ways to quantize the reward score to leverage it as valid feedback when training the model.

## 7 ETHICS STATEMENT

Our work on improving figure caption generation is important in building accessible assistive tools for scientific community and visually impaired people. However, like many works in the area of generative AI, our work/general ideas also carry the risk of misuse i.e. our proposed method can be advertised by a third party as a deployable product, when in fact, we believe that our proposed method is a research endeavor and still has room for improvement. Another potential negative impact of our work could be the complacent consideration of generating human feedback without due consideration to human subjects involved. This is our key motivation to make our dataset with feedback labels public, to allow interested researchers to develop and benchmark their own methods that require feedback.

Finally, we comment on the dataset privacy considerations for the proposed benchmark. Our proposed dataset and other datasets considered in this work are licensed for academic/non-commercial research (Creative Commons Attribution-Non Commercial-Share Alike 4.0 International License). Our proposed dataset does not contain any personal information.

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

APPENDIX

# A OVERVIEW

In the following subsections,

- We provide details of our quality metrics used for evaluating a figure-caption pair, our experimental setup, baseline model details and a discussion on the qualitative comparative results.
- Following the guidelines mentioned in Gebru et al. (2021), we provide information regarding data composition, data collection procedure, use cases for our dataset. The document also includes Author statement, Licensing and Maintenance Plan.

Our dataset along with its documentation and code has been made publicly available at:

**Benchmark:** `https://figshare.com/s/c034fd77bea9475319cb`
**Code:** `https://github.com/FigCapsHF/FigCapsHF`
**Documentation:** `https://figcapshf.github.io/`

## A.1 DESCRIPTION OF METRICS USED FOR FEEDBACK ASSESSMENT

We followed Huang et al. (2023) to evaluate a given figure-caption pair from the perspective of a reader. Specifically, we used the following measures:

- **Helpfulness:** This is a subjective measure to evaluate whether a given caption is able to inform the reader about the information conveyed in the corresponding figure.
- **Takeaway:** This measure is used to assess a given caption based on whether it is able to convey a conclusive information about the given figure image.
- **Visual-descriptiveness (visual):** We define visual descriptiveness of a given caption as a measure of how much the given caption is grounded with respect to the figure. For example, a caption that describes the visual elements of the figure like color and shape should be more informative to the readers.

- **Image-text (OCR):** We formulate OCR as a metric to evaluate if the given caption included textual elements of the figure like title, legends and labels when describing the figure.

## A.2 EXPERIMENTAL SETUP

### A.2.1 DATASETS

For all our models, we use the same splits in our benchmark dataset; this portion contains 106,834 training pairs, 13,354 validation pairs, and 13,355 test pairs. The primary difference between our baseline and RLHF models is the human-feedback augmented figure-captions that are used for training the latter (figure-images remain the same) and testing figure-caption pairs remain the same for both.

**Annotation details of Human-Feedback set**: We selected the annotators based on their expertise in the areas of computer vision/natural language processing and machine learning. Our annotator pool consisted of 10 Ph.D. graduates and active graduate students (no authors) with published work in the CV, NLP, and ML conferences. We randomly selected 438 figure-caption pairs from the dataset to be annotated. Each annotator was provided 2 weeks time to annotate the data subset. For each sample, annotators were asked to provide ratings on a five-point Likert scale for the following attributes [OCR, Visual, Takeaway, Helpfulness]. For each sample, the following descriptions were provided:

- OCR: The caption includes named entities or important words/numbers in the figure(e.g., title, legends, labels, etc.).
- Visual-Descriptiveness: The caption includes some visual characteristics of the figure (e.g., color, shape, trend, etc.).
- Takeaway: The given caption explicitly states the high-level takeaway message or the conclusion that the figure attempted to convey.
- Helpfulness: The caption was helpful in understanding the message that the figure is attempting to convey.

**Human-Feedback Augmented Caption** For our RLHF-trained models, we generate human-feedback augmented figure-captions to align the model to human preferences. In this process, for each caption, we first use MCSE (Zhang et al., 2022) to generate text-embeddings for the captions in the human annotated dataset ( 400 pairs). An auxiliary scoring-model (MLP Regressor) is then trained to predict the reader-preference scores using these embeddings, and later used to predict human feedback scores for the entire dataset; we pick the median of these scores as a pivot and label all captions with higher scores as "good", and lower scores as "bad". After pre-pending our captions with these annotations, we effectively train our models in a UDRL framework. Code to implement and generate new human-feedback augmented captions are provided in the GitHub repository.

### A.2.2 EVALUATION METRICS

We evaluate the generated captions using a variety of common metrics. **ROUGE-L** (Lin, 2004) is a recall-oriented metric which uses the Longest Common Subsequence between the reference and the model generated caption, we report the F1 score. **BLEU** (Papineni et al., 2002) is a precision-oriented metric which uses n-gram overlap, and an additional penalty for sentence brevity. Here, we are using **BLEU@4** (i.e $n = 4$ for n-gram overlap) **METEOR** (Banerjee & Lavie, 2005) measures generalized unigram-overlap and computes a combination of the precision and recall. For a summary of the evaluation metrics leveraged by traditional image captioning works, see Stefanini et al. (2022).

### A.2.3 BASELINES

For comparative evaluation of our proposed framework, we selected methods based on the information used to generate a caption. Specifically, we categorize the baselines models into following categories:

- **Figure-only:** We refer to a method as 'Figure-only' if the given method computes an output text based on uni-modal embedding of the input image. Model architecture under this category generally comprises of some combination of a vision encoder and a text decoder module.

- **OCR-only:** Similar to above, if a method generates an output text using only text as input to the text decoder model, we classify the same as 'Text-only' methods. Specific to our case, we can extract some textual descriptions of a given figure by applying an off-the-shelf OCR method. Hence from here on, we explicitly refer to methods falling under the above-mentioned criteria as 'OCR-only' models. Methods under this category utilizes a text encoder and text decoder modules as part of their model architecture.

- **Figure-Caption:** Finally for methods which compute multi-modal embedding from text and image uni-modal embeddings to be utilized for generating output text using a text decoder, we categorize them as 'Figure-Caption' methods. All the methods under this category generally include a vision encoder, text encoder and text decoder modules as part of their model architecture.

We evaluate a variety of strong image-captioning models and a text-summarization model as our baselines. We provide details of individual models below:

**Unimodal Vision-Encoder Language-Decoder Models**. These models consist of a pre-trained Vision-Encoder (e.g. BEiT (Bao et al., 2022), ViT (Dosovitskiy et al., 2021)) and a pre-trained Text-Decoder/Language model (e.g. GPT-2 (Radford et al., 2019), RoBERTA (Liu et al., 2019)). The two submodules are not pre-trained jointly, and only aligned during fine-tuning via randomly initialized cross-attention layers in the decoder. These models simply take in the figure-image and generate the corresponding caption.

**Pegasus** (Zhang et al., 2020) is a Transformer-based pre-trained model for text-summarization. We use PEGASUS to generate figure-captions by summarizing the OCR extracted from the image itself.

**TrOCR** (Li et al., 2021) is a Transformer-based OCR model designed to extract text from a given image. It uses BEiT/DEiT as a vision encoder and RoBERTA as a text decoder, similar to the aforementioned image-to-text models, with the addition of an OCR-focused pre-training. We fine-tuned the model to generate a caption from a given figure-image.

**GIT** (Wang et al., 2022a) is a Generative Image-to-Text model. It uses a pre-trained Vision-Transformer encoder and a randomly initialized Language Transformer decoder (e.g. BERT(Devlin et al., 2018)), similar to the aforementioned image-to-text models, and further jointly pre-trains them using the Language Modeling task. We evaluated the performance of both fine-tuned and pre-trained versions of GIT.

**BLIP** (Li et al., 2022a) is a Multi-Modal Vision-Language decoder model. It has a similar architecture to the Vision-Encoder Decoder image-to-text models, but utilizes interchangeable attention layers in the text-decoder to behave as either an unimodal encoder, an image-grounded text encoder or an image-grounded text decoder. The model is pre-trained using the LM, ITM and ITC losses jointly.

**PromptCap**(Hu et al., 2022) is a prompt-based image-captioning model. In addition to taking an image, the model can also incorporates a user-defined prompt to guide the generated caption. PromptCap utilizes a pre-trained Transformer-based encoder-decoder model, namely OFA (Wang et al., 2022b) which is further pre-trained. PromptCap is evaluated zero-shot using its pre-trained version due to lack of available documentation.

**Flamingo-mini** (Alayrac et al., 2022) is a Transformer-based encoder-decoder model which has a similar structure to the aforementioned image-to-text models. However, the pre-trained vision encoder and text decoder are frozen and an additional module is used to learn transformed visual representations for the frozen language model to attend to.

**CLIPCap** (Mokady et al., 2021) is a Transformer-based encoder-decoder model. It utilizes CLIP as an image encoder, and using a mapping network, maps image embeddings to a prefix which is used by a text-decoder, namely GPT2, to generate a caption. The pre-trained modules and the freshly-initialized mapping network are simply fine-tuned during the training process.

From the set of baseline models described above, we fine-tuned ViT+RoBERTA, ViT+GPT2, BEiT+GPT2, GIT, BLIP and CLIPCap on the training set of our dataset. To understand zero-shot performance for figure-captioning task, we evaluated Pegasus, TrOCR, PromptCap and Flamingo-mini models by using their pretrained weights for inference without fine-tuning them on our dataset.

For all fine-tuning experiments, we used AdamW optimizer with $\beta_1 = 0.9$ & $\beta_2 = 0.99$. We fine-tuned ViT+RoBERTA, ViT+GPT2, BEiT+GPT2 for 5 epochs with batch size 8. We used a linear rate scheduler with an initial learning rate of $2e - 5$; generation was handled using a greedy strategy. For training GIT, BLIP and CLIPCap models, we used a learning rate of $1e - 5$ and used nucleus sampling for text generation during inference.

### A.3 QUALITATIVE ANALYSIS

In this section, we provide a detailed qualitative analysis of the output of BLIP-RLHF and BLIP (Fine-tuned) models.

**Comparative analysis:** In the first example shown at the top left in Figure 3, we see that the generated caption with the base model BLIP has many issues. For instance, it seems to have identified the word "edges" from the name of the model "Deep-Edge" used in the figure, despite that the figure does not actually show the number of edges in each experiment as the caption mentions. Instead, it shows the average epoch time in seconds for each of the different experiments, which is roughly captured by the BLIP-RLHF caption. In the second example shown in the middle of Figure 3, the BLIP model completely hallucinates the caption whereas the BLIP-RLHF caption reveals the essence of the figure while also seemingly using the semantics of this specific chart-type, *e.g.*, the phylogenetic tree shows the evolutionary relationships between different groups of fish and from the phylogenetic tree we can see how large each group is and the similarities between the groups of fish as well. This also illustrates the ability of our approach to generalize to a variety of different chart types as we only obtained actual human feedback for line charts. For the captions generated for the chart shown at the right in Figure 3, we see that BLIP generates a completely useless caption that has no alignment with the actual chart. In comparison, the caption generated using BLIP-RLHF mentions the estimated and actual curves present in the chart while also correctly indicating that these curves are plotted in terms of time. Most strikingly, the generated caption refers to the curves using their color (*i.e.*, red line, blue dots), hence, the generated caption not only mentions important text from the chart, but also refers to the visual properties of the curves when mentioning them in the generated caption.

**Human-Evaluation of model generated captions**: To further evaluate the generated captions, we conducted a small-scale human evaluation experiment. Specifically, we randomly select 100 figures from the Test set of our proposed benchmark and generate captions using the BLIP and BLIP-RLHF models. We present the triplet of Figure, corresponding BLIP, and BLIP-RLHF generated captions (after randomizing the order of the two captions) to 10 human subjects. Each human subject is asked to rank the two captions based on which caption they think is better. We ask the subjects to specifically consider helpfulness, visual-descriptiveness, OCR alignment, and takeaway while ranking individual pairs of captions. To guide the subjects, we first explain each metric [helpfulness, visual-descriptiveness, OCR alignment, and takeaway] and present each human subject with 100 samples from our human-annotated dataset with individual figures, ground truth caption, and the corresponding metric scores (recorded in 5-point Likert scale). From our study, we find that on average 85% of the time, BLIP-RLHF generated caption was selected as the better caption relative to BLIP generated caption. From our small-scale study, we conclude that RLHF does improve the quality of the captions when compared to fine-tuning existing Vision Language models for the task of figure-caption generation.

## B DATASHEET

### B.1 MOTIVATION

**For what purpose was the dataset created?** We created this dataset to provide researchers ability to develop and evaluate their respective figure-to-caption generation pipelines for reader preference-aligned caption generation.

**Who created the dataset (e.g., which team, research group) and on behalf of which entity(e.g., company, institution, organization)?** We would provide the details of the authors upon acceptance of the paper, due to double-blind review process.

**Who funded the creation of the dataset?** No funding was received in any form in the creation of this dataset.

### B.1.1 AUTHOR STATEMENT

The authors of this paper bear all responsibilities for the distribution, and maintenance of our proposed dataset. This document follows the Datasheet format (Gebru et al., 2021) whenever applicable.

## B.2 DISTRIBUTION

**Will the dataset be distributed to third parties outside of the entity (e.g., company, institution, organization) on behalf of which the dataset was created?** Yes, the dataset is public and available for usage on the internet.

**How will the dataset will be distributed (e.g., tarball on website, API, GitHub)?** The dataset and the corresponding codebase used in generating the dataset is available through the following links:

**Benchmark:** `https://doi.org/10.6084/m9.figshare.23504517`
**Code:** `https://github.com/FigCapsHF/FigCapsHF`
**Documentation:** `https://figcapshf.github.io/`

**Have any third parties imposed IP-based or other restrictions on the data associated with the instances?** No.

**Do any export controls or other regulatory restrictions apply to the dataset or to individual instances?** No.

## B.3 MAINTENANCE

**Who will be supporting/hosting/maintaining the dataset?** The authors will be supporting, hosting and maintaining the dataset.

**How can the owner/curator/manager of the dataset be contacted (e.g., email address)?** We would provide the details of the contact persons upon acceptance of the paper, due to double-blind review process.

**Is there an erratum?** No. We will accordingly make announcements if there is any.

**Will the dataset be updated (e.g., to correct labeling errors, add new instances, delete instances)?** Yes. Announcements regarding any updates to the dataset and code will be posted here: `https://github.com/FigCapsHF/FigCapsHF`

**If the dataset relates to people, are there applicable limits on the retention of the data associated with the instances (e.g., were the individuals in question told that their data would be retained for a fixed period of time and then deleted)?** N/A

**Will older versions of the dataset continue to be supported/hosted/maintained?** Yes.

**If others want to extend/augment/build on/contribute to the dataset, is there a mechanism for them to do so?** Yes.

## B.4 COMPOSITION

**What do the instances that comprise the dataset represent?** Please refer to section B.7 for a detailed description of the dataset composition.

**How many instances are there in total (of each type, if appropriate)?** in total we have 06,834 training pairs, 13,354 validation pairs, and 13,355 test figure-caption pairs with feedback scores.

**Does the dataset contain all possible instances or is it a sample (not necessarily random) of instances from a larger set?** The dataset contains all possible instances.

**Is there a label or target associated with each instance?** Yes. Each figure image in the dataset has a corresponding caption and a set of values representing the predicted feedback score for metrics ('helpfulness', 'ocr', 'visual', 'takeaway'.

**Is any information missing from individual instances?** No.

**Are relationships between individual instances made explicit (e.g., users' movie ratings, social network links)?** N/A

**Are there recommended data splits (e.g., training, development/validation, testing)?** Yes. The dataset consists of 3 splits: Train, Validation and Test. We have explicitly provided individual splits as separate data folders.

**Are there any errors, sources of noise, or redundancies in the dataset?** No.

**Is the dataset self-contained, or does it link to or otherwise rely on external resources (e.g., websites, tweets, other datasets)?** The dataset is entirely self-contained and does not require any external resources.

**Does the dataset contain data that might be considered confidential?** No.

**Does the dataset contain data that, if viewed directly, might be offensive, insulting, threatening, or might otherwise cause anxiety?** No.

### B.5 COLLECTION PROCESS

**Who was involved in the data collection process (e.g., students, crowdworkers, contractors) and how were they compensated (e.g., how much were crowdworkers paid)?** The authors were involved in the curation of the data obtained from a publicly available source.

**Over what timeframe was the data collected?** February 2023-May 2023

### B.6 USES

**Has the dataset been used for any tasks already?** Our work on human feedback aligned figure caption generation uses the proposed dataset.

**Is there a repository that links to any or all papers or systems that use the dataset?** N/A

**What (other) tasks could the dataset be used for?** Evaluating image-to-text generation models for a domain specific performance.

**Is there anything about the composition of the dataset or the way it was collected and preprocessed/cleaned/labeled that might impact future uses?** No.

### B.7 DATA FORMAT

For each figure-caption pair, the figure-image is stored as a PNG, and the figure-caption (with associated metadata) is stored in a JSON format. 4 is an example from the dataset.

In each figure-caption's metadata file, the fields are:

- **contains-subfigure:** boolean (if figure-image contains subfigures)
- **paper-ID:** the unique paper ID in the arXiv dataset
- **figure-ID:** the extracted figure ID of paper (the index is not the same as the label in the caption)
- **figure-type:** the figure type
- **0-originally-extracted:** original figure-caption
  - **caption:** caption after each normalization
  - **sentence:** a list of segmented sentences
  - **token:** a list of tokenized words
- **1-lowercase-and-token-and-remove-figure-index:** Removed figure index and the captions in lowercase
  - Same substructure as 0-originally-extracted
- **2-normalized:**
  - **2-1-basic-num:** caption after replacing the number

- * Same substructure as 0-originally-extracted
  - **2-2-advanced-euqation-bracket:** caption after replacing the equations and contents in the bracket
    * Same substructure as 0-originally-extracted
- **Img-text:** texts extracted from the figure, such as the texts for labels, legends ... etc.

Within the "human-feedback" field, we have the inferred human-feedback for the different metrics (helpfulness, ocr, takeaway, and visual). The tokens are decided based on the median score of the dataset on that metric.

- **Helpfulness:** Expert's rating on how helpful a caption is to understand a scientific figure
  - **Score:** predicted score
  - **Token:** [Good]/[Bad]
  - **caption-prepend:** 1-lowercase-and-token-and-remove-figure-index caption with the token
- **Takeaway:** Expert's rating on the takeaway from the scientific image
  - Same substructure as Helpfulness
- **OCR:** Expert's rating on the OCRs expressiveness
  - Same substructure as Helpfulness
- **Visual:** Expert's rating on the visualness of the scientific figure
  - Same substructure as Helpfulness

```
1   {
2       "contains-subfigure": false,
3       "Img-text": ["Attack", "duration", "[s]", "350", "300", ...],
4       "paper-ID": "1001.0025v1",
5       "figure-ID": "1001.0025v1-Figure2-1.png",
6       "figure-type": "Graph Plot",
7       "human-feedback":{
8           "helpfulness": {
9               "score": 4.27,
10              "label": "GOOD",
11              "caption-prepend": "[GOOD] impact of the replay ...",
12          },
13          "ocr": {
14              "score": 4.19,
15              "label": "GOOD",
16              "caption-prepend": "[GOOD] impact of the replay ...",
17          },
18          "visual": {
19              "score": 2.86,
20              "label": "BAD",
21              "caption-prepend": "[BAD] impact of the replay ...",
22          },
23          "takeaway": {
24              "score": 4.7,
25              "label": "GOOD",
26              "caption-prepend": "[GOOD] impact of the replay ...",
27          },
28      }
29      "0-originally-extracted": "Figure 2: Impact of the replay ...",
30      "1-lowercase-and-token-and-remove-figure-index": {
31          "caption": "impact of the replay attack, as a function ...",
32          "sentence": ["impact of the replay attack, as a ..."],
33          "token": ["impact", "of", "the", "replay", "attack", "..."]
34      },
35      ...
36  }
```

Figure 4: Human Feedback Benchmark Data Example for Figure-Caption Generation with RLHF

### B.7.1 READING DATA

For all figure-caption pairs, all of the figure-images are in their respective train/val/test subfolders under the "No-Subfig-Img" folder. The corresponding figure-captions and associated metadata are in

their respective train/val/test subfolders under the "Caption-All' folder, bearing the same filename as their image. In order to read the data, one can read the file-names of all the figure-images in a particular data-split, and retrieve the corresponding figure-caption metadata using the image file-names (instead iterating through the captions also works). Another approach is to iterate through the "file_idx.json" file under the "List-of-Files-for-Each-Experiments/First-Sentence/(train/val/test)" folder, which contains a list of all image-names we used for that data split.

### B.7.2 REPRODUCIBILITY

We have provided easy access to the benchmark dataset which was used to conduct all of our experiments, including the augmented caption that was used during RLHF fine-tuning.

We have also provided access to a github repository, which contains the code used to train a baseline, fine-tune a model using human-feedback, and evaluate the model on the test set.

