# OpenReview forum: "FigCaps-HF: A Figure-to-Caption Generative Framework and Benchmark with Human Feedback"
_ICLR.cc/2025/Conference — ICLR 2025 Conference Withdrawn Submission_

### Official Review · Reviewer_miB8 · 2024-10-28

**Soundness:** 2
**Presentation:** 2
**Contribution:** 2
**Rating:** 3
**Confidence:** 4

**Summary:**

This paper investigates the figure-to-caption generation problem. The key idea is to train a feedback model on human experts' feedback and apply it to large-scale unlabeled samples. This leads to a benchmark FigCaps-HF with annotated reward scores.
The reward signal is transformed into a control token `<GOOD>` or `<BAD>` to inform the model of the quality of the caption during the learning process.
Experiments with image-to-text models such as BLIP & ViT+GPT2 show that incorporating the feedback signal is beneficial regarding the generation metrics. Further analysis examines various aspects of the proposal such as the effect of feedback from different perspectives and reward score generalizability to OOD distributions.

**Strengths:**

- Figure-to-caption generation is an interesting question and of great potential for aiding scientific paper writing;
- The idea of incorporating expert feedback into figure-to-caption intuitive and effective as demonstrated by the experiments.

**Weaknesses:**

While the explorations and efforts of this paper is greatly appreciated, I think this paper can be improved quite a lot by addressing the following concerns.

- Old backbone used: Given the rapid evolution of large vision-language models (e.g., LLaVA-series,  Qwen-VL)., they have shown promising results regarding various benchmarks including figure-to-caption [1]. I would suggest the authors incorporate the results of commercial models to serve as an indicator for the sota performance and perform experiments (i.e., reward-guided tuning) with open-sourced LVLMs. BLIP (2021) & ViT+GPT2 (2019) are not sufficient as they are using (relatively) small LLMs, where I believe the rich knowledge in recent LLMs would help a lot for producing high-quality captions.

- Develop better metrics for scientific figures captioning: In current Table 2, the BLEU scores improve from 0.01x to 0.02, rendering the results less convincing.  I understand that such an open-ended generation is hard to evaluate.  N-gram overlapping-based metrics such as ROUGE-L and BLEU are insufficient to comprehensively assess the caption. Sentence-embedding-based metrics such as Senetence-BERT and LLM-aided evaluation with the golden captions would be helpful for this scenario.

- The baseline methods are insufficient: Using the reward model to annotate `<GOOD>` and `<BAD>` indicators and then adopting it into training is common practice in RLAIF papers nowadays. I would like the author to compare the method with PPO method and unlikelihood training of `<BAD>` captions to provide more insights and justify the experimental choice better.

- Benchmark quality check: The paper claims to propose a benchmark annotated by a reward model. However, to serve as a benchmark, rigorous human check is required to ensure the quality. I would like to see a consistency check of a subset and the reward model accuracy on the held-out set.



[1] Multimodal ArXiv: A Dataset for Improving Scientific Comprehension of Large Vision-Language Models, ACL 2024

**Questions:**

-  How did your trained reward model perform on the validation set?

##  Layout
There is an empty line above Table 7. I would recommend the author place tables properly  (e.g., at the top of each page) and merge short tables for better visibility.

---

### Official Review · Reviewer_eE1e · 2024-11-02

**Soundness:** 3
**Presentation:** 3
**Contribution:** 3
**Rating:** 6
**Confidence:** 3

**Summary:**

This paper introduces an RLHF benchmark and framework for generating captions from figures using limited human feedback. The approach involves learning an oracle model from a small amount of feedback to predict human scores for new, unseen figure-caption pairs. Extensive experiments demonstrate the method's effectiveness, and the authors provide a comprehensive benchmark dataset to support future research in figure-to-caption generation via RLHF.

**Strengths:**

**Incorporation of Domain Expert Feedback**: The FigCaps-HF framework can incorporate domain expert feedback to optimize generated captions, making them more suitable for readers' needs.

**Combination of Automatic Evaluation and RLHF**: The framework includes an automatic method for evaluating the quality of figure-caption pairs and utilizes a reinforcement learning with human feedback (RLHF) approach to further optimize the generative model.

**Release of a Large-Scale Benchmark Dataset**: A large-scale benchmark dataset on figure-caption pairs is provided to support further evaluation and development of RLHF techniques for this problem.

**Weaknesses:**

**Layout and Aesthetic Issues**: The layout of the paper could be improved, and the figures are not visually appealing, which affects the overall presentation and readability.

**Weak Baseline Models**: The paper uses BLIP and ViT+GPT2 as baseline models, which are relatively weak. Given the rapid advancement of multimodal large models, experiments should be conducted on more and stronger baselines to comprehensively validate the effectiveness of the proposed approach.

**Questions:**

In Figures 2 and 3, the authors mainly illustrate their points through a few examples, which is relatively weak since there are always some examples that meet the requirements. Could the authors provide some statistical evidence to support these claims or use other methods to prove their points from different perspectives?

---

### Official Review · Reviewer_uWws · 2024-11-03

**Soundness:** 2
**Presentation:** 3
**Contribution:** 2
**Rating:** 3
**Confidence:** 3

**Summary:**

The paper presents FigCaps-HF, a new framework for figure to caption generation. Specifically, the paper incorporates human feedback through a RLHF setup to improve the captioning abilities of image-to-text models. The work also introduces a benchmark, as part of this framework to evaluate the quality of figure-caption pairs by training on a small subset of human-annotated data. Experiments on metrics such as BLEU and ROUGE are performed to demonstrate the effectiveness of the approach, with ablations performed validating the varying components of the method.

**Strengths:**

1. The paper is well written and easy to follow.
2. I appreciate the release of the dataset and documentation; which are a good contribution.
3. The conducted experiments and ablation, within the scope of the paper, are well motivated and justify the presented approach.

**Weaknesses:**

1. The presented baselines are missing some key models -- there are many new models such as LLaVA-NeXT that should be used as baselines. Guaranteed, they are larger models but in general does the zero-shot performance of these multimodal LLMs compare against the presented method.
2. The RLHF component should also be compared against existing methods such as direct preference optimization (DPO) to fully understand the advantages over the human feedback component.
3. There are no experiments demonstrating the zero-shot performance of the proposed method to other tasks. For example, what is the zero-shot performance on baseline vs proposed method on table captioning?

**Questions:**

1. What are the reference captions used for computing ROUGE, BLEU scores etc.?
2. Are 4 different models being trained to generate the 4 respective scores? If yes, is this not computationally expensive and can this not be regressed by 1 model?
3. _Generalizability of the human feedback prediction model_ : I am not convinced that is proof of generalization. The validation set consists of the same 438 human annotated samples, which the model is trained to optimize.
4. Since the "core" idea depends on human annotations, there should be a brief discussion about the demographics in the main paper.

---

### Official Review · Reviewer_UxAK · 2024-11-03

**Soundness:** 2
**Presentation:** 3
**Contribution:** 2
**Rating:** 3
**Confidence:** 4

**Summary:**

This paper focuses on the figure-to-caption generation task, introducing a new framework that incorporates domain expert feedback for generating and evaluating captions. A lightweight reward model is trained on a small set of figure-caption pairs and used to score other pairs, with reinforcement learning employed to train the caption model for improved performance. Experimental results show that the RL approach can enhance performance.

However, several issues are noted, including the lack of discussion on related work and the absence of evaluations using current state-of-the-art multimodal language models, such as proprietary models like GPT-4, Claude 3.5, and Gemini, as well as open-source models like Qwen-2 VL, InternVL, and IDEFICS. Additionally, it is unclear how the learned reward model sustains strong performance given the limited training data, and there is insufficient detail regarding design choices and comparisons involving vision-language models for rewards.

**Strengths:**

1. This work introduces a framework that incorporates domain expert feedback for figure caption generation, an area that is underexplored.
2. The lightweight reward model is efficiently trained on a small dataset to score figure-caption pairs, yielding impressive results as demonstrated in Table 1.
3. The reinforcement learning approach effectively enhances the caption model's performance.

**Weaknesses:**

1. The paper lacks discussion of related work, including notable studies such as ArxivCap [1] and MMSci [2].
2. There is no evaluation of current state-of-the-art multimodal language models, including proprietary models like GPT-4, Claude 3.5, and Gemini, or open-source models like Llava, Qwen-2 VL, InternVL, and MiniCPM.
3. It is unclear how the learned reward model ensures robust performance given the limited training data. Additionally, there is insufficient detail on the design choices and ablation studies related to using a vision-language model for rewards.
4. The ROUGE scores might not be good metrics for evaluating caption. Some other better metrics such as FactScore [3] should be considered.

[1] Li, Lei, et al. "Multimodal arxiv: A dataset for improving scientific comprehension of large vision-language models." arXiv preprint arXiv:2403.00231 (2024).
[2] Li, Zekun, et al. "Mmsci: A multimodal multi-discipline dataset for phd-level scientific comprehension." arXiv preprint arXiv:2407.04903 (2024).
[3] Min, Sewon, et al. "Factscore: Fine-grained atomic evaluation of factual precision in long form text generation." arXiv preprint arXiv:2305.14251 (2023).

**Questions:**

1. Why not train a vision-language model to evaluate image-caption pairs, instead of using a regression function over fixed embedding functions? Additionally, how was the embedding function selected?
2. What is the performance of state-of-the-art multimodal language models, including proprietary models such as GPT-4, Claude 3.5, and Gemini, as well as open-source models like Llava, Qwen-2 VL, InternVL, and MiniCPM?
3. The evaluation uses metrics (ROUGE) that have been shown to poorly measure quality of LLM-generated text. Non-overlapped based metrics, like FactScore would be more appropriate.

---

### Note · Authors · 2024-12-03

**Comment:**

We thank the reviewers for their helpful feedback. We have decided to withdraw our work from the conference based on the responses.

**Withdrawal Confirmation:**

I have read and agree with the venue's withdrawal policy on behalf of myself and my co-authors.